# TUT4CRS: Time-aware User-preference Tracking for Conversational Recommendation System

Dongxiao He
Tianjin Key Laboratory of
Cognitive Computing and
Application, College of
Intelligence and Computing
Tianjin University
Tianjin, China
hedongxiao@tju.edu.cn

Jinghan Zhang
Tianjin Key Laboratory of
Cognitive Computing and
Application, College of
Intelligence and Computing
Tianjin University
Tianjin, China
zhangjh0621@tju.edu.cn

Xiaobao Wang*
Tianjin Key Laboratory of
Cognitive Computing and
Application, College of
Intelligence and Computing
Tianjin University
Tianjin, China
wangxiaobao@tju.edu.cn

Meng Ge
Saw Swee Hock School of
Public Health
National University of
Singapore
Singapore, Singapore
gemeng@tju.edu.cn

Zhiyong Feng
Tianjin Key Laboratory of
Cognitive Computing and
Application, College of
Intelligence and Computing
Tianjin University
Tianjin, China
zyfeng@tju.edu.cn

Longbiao Wang
Tianjin Key Laboratory of
Cognitive Computing and
Application, College of
Intelligence and Computing
Tianjin University
Tianjin, China
longbiao_wang@tju.edu.cn

Xiaoke Ma
School of Computer
Science and Technology
Xidian University
Xi'an, China
xkma@xidian.edu.cn

## Abstract

The Conversational Recommendation System (CRS) aims to capture user dynamic preferences and provide item recommendations based on multi-turn conversations. However, effectively modeling these dynamic preferences faces challenges due to conversational limitations, which mainly manifests as limited turns in a conversation (quantity aspect) and low compliance with queries (quality aspect). Previous studies often address these challenges in isolation, overlooking their interconnected nature. The fundamental issue underlying both problems lies in the potential abrupt changes in user preferences, to which CRS may not respond promptly. We acknowledge that user preferences are influenced by temporal factors, serving as a bridge between conversation quantity and quality. Therefore, we propose a more comprehensive CRS framework called Time-aware User-preference Tracking for Conversational Recommendation System (TUT4CRS), leveraging time dynamics to tackle both issues simultaneously. Specifically, we construct a global time interaction graph to incorporate rich external information and establish a local time-aware weight graph based on this information to adeptly select queries and effectively model user dynamic preferences. Extensive experiments on two real-world datasets validate that TUT4CRS can significantly improve recommendation performance while reducing the number of conversation turns.

*Corresponding Author

## CCS Concepts

• **Information systems** → **Recommender systems**; • **Mathematics of computing** → *Graph algorithms*.

## Keywords

Conversational Recommendation System, Graph Neural Networks, Graph Representation Learning

**ACM Reference Format:**
Dongxiao He, Jinghan Zhang, Xiaobao Wang, Meng Ge, Zhiyong Feng, Longbiao Wang, and Xiaoke Ma. 2024. TUT4CRS: Time-aware User-preference Tracking for Conversational Recommendation System. In *Proceedings of the 32nd ACM International Conference on Multimedia (MM '24), October 28-November 1, 2024, Melbourne, VIC, Australia.* ACM, New York, NY, USA, 9 pages. https://doi.org/10.1145/3664647.3681259

## 1 Introduction

The recommendation system plays a crucial role in efficiently delivering personalized recommendations to users based on their interests [31]. By modeling user preferences, these systems effectively address the problem of information overload [24]. However, traditional recommendation systems rely on the user's historical interactions to model their interests, having the assumption that these interests remain static over time [4]. Consequently, it is challenging to capture fine-grained and dynamic user preferences accurately.

Conversational recommendation systems (CRS) [17] are designed to capture user dynamic preferences in a more fine-grained manner by allowing the system to directly ask users for their preferences and gather feedback. CRS can be divided into two scenarios: single-turn and multi-turn conversational recommendation. In the single-turn conversational recommendation (SCR), the system stops after recommending items to the user, regardless of whether the user is satisfied or not. On the other hand, the multi-turn conversational recommendation (MCR) continues to interact with the user by

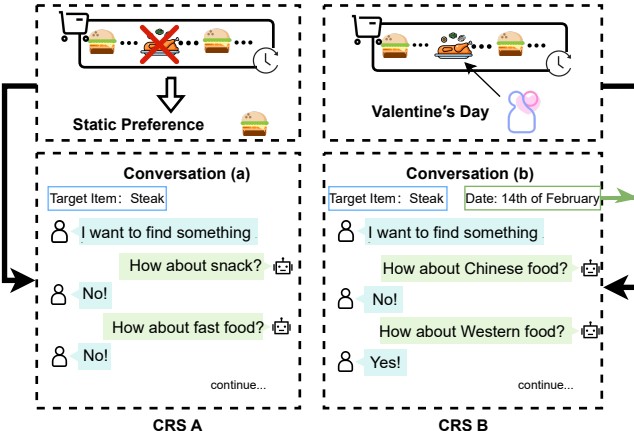

**Figure 1: Illustration of two types of CRS.**

asking questions or making recommendations until it successfully meets the user's needs or reaches a maximum number of turns [5, 18]. For a more interactive and personalized recommendation experience, MCR aligns better with real-world requirements, allowing for ongoing interaction with users and ensuring that their needs are fully addressed.

Existing methods in multi-turn conversational recommendation (MCR) typically rely on direct user input to model their preferences, known as user dynamic preferences. These preferences are then used to provide personalized recommendations. However, the effectiveness of these methods in achieving their intended outcomes is limited due to the issue of conversation limitation. This limitation manifests in two distinct ways: limited turns in a conversation (quantity aspect) and low compliance of queries (quality aspect). The first aspect, limited turns in a conversation, refers to the constraint on the number of turns allowed in a conversation. This limitation poses a challenge as it hinders the system's ability to gather sufficient information and accurately model user dynamic preferences within the given timeframe. The second aspect, low compliance of queries, pertains to the difficulty in selecting the most appropriate query to ask the user in each turn. With a large set of query candidates to choose from, it becomes increasingly challenging for the system to accurately identify the correct query.

To address the issue of quantity, some works [9, 33] have attempted to obtain additional information beyond conversational information and mine the user's cyclical behavior through the user's historical interactions to model user static preferences. When there is no clear dynamic preference in the user's conversation, recommendations can be made using the user static preference. Regarding the issue of quality, some works [18, 34] model CRS as interactive path reasoning on a graph, constraining the candidate query set to only include the neighbor nodes of the current node, thereby greatly reducing the candidate space and selecting the correct query.

Previous research has often addressed these two issues in isolation, neglecting their intrinsic interconnections, which can lead to various challenges. As depicted in Figure 1, the left side illustrates that most existing methods typically tackle the quantity aspect by modeling user static preferences (i.e., represented as a "hamburger"), achieved by filtering out occasional behaviors from the user order

history. Consequently, the agent's query selection tends to prioritize static preferences. However, inaccurate additional information may result in inappropriate queries, rendering the candidate set unfilterable, exacerbating the quality aspect problem. Thus, these two issues are interrelated and mutually influence each other. It is imperative to address them jointly to devise a comprehensive solution. Fortunately, user behavior often demonstrates temporal influences, reflecting differences in users' consumption habits at different times. This is exemplified on the right side of Figure 1, where the occasional behavior of ordering roast duck in the user's historical orders is attributed to Valentine's Day, reflecting users' tendency to select more formal and expensive meals on such occasions. Considering time enables us to acquire accurate external information to tackle the quantity aspect problem while aiding in candidate set filtration. Additionally, the attributes within the candidate set inherently possess temporal properties, allowing us to further refine the candidate set using time information to address the quality aspect problem. Motivated by this observation, we propose utilizing the time factor as a means to address both problems simultaneously, serving as a bridge between them.

However, incorporating time factors to tackle the aforementioned challenges presents significant difficulties. When integrating time factors into the fine-grained modeling of user static preferences, occasional yet significant behaviors may inadvertently be filtered out as noise. Moreover, a vast number of candidate items and attributes can diminish the efficiency and accuracy of recommendations and queries. Addressing how to leverage time factors for effective filtering remains a formidable challenge—specifically, determining how to efficiently discern the relationships among time, items, and attributes, and subsequently filter candidate sets based on these relationships. Most importantly, achieving integration between these two modules in a mutually reinforcing manner poses a critical challenge.

To solve these problems, we propose a novel framework, i.e., **T**ime-aware **U**ser-preference **T**racking for **C**onversational **R**ecommendation **S**ystem (TUT4CRS), which consists of two main components. Firstly, we utilize user historical interactions to build a Global Time Interaction Graph. This graph serves as a bridge between the user and the item, allowing us to model the user static preferences. We can learn about the user's consumption habits from their occasional behaviors. The item can also gather information about its implied temporal features. This approach enables us to capture both the user long-term preferences and short-term preferences. Secondly, we construct a Local Time Aware Weight Graph to leverage the previously learned information. This graph dynamically learns the user's perception of time and helps us make query selections based on this understanding. By incorporating this component, we can efficiently model the user's finer-grained dynamic preferences.

To summarize, the contributions of this paper include:

- We acknowledge the temporal impact on user behavior and propose linking the quality and quantity aspects of conversational recommendation through the time factor, opening a new direction for future research.
- We propose a novel framework that utilizes time factor to capture incidental yet significant behaviors beyond users' dynamic preferences. This framework integrates this information to aid query selection during conversations, leading

to a more precise depiction of user dynamic preferences in a shorter number of turns.

- We conduct experiments on two real-world datasets and demonstrate that TUT4CRS can effectively improve the performance of conversational recommendations.

## 2 Related Works

Unlike traditional recommendation systems [2, 7, 8], Conversational Recommendation Systems (CRS) aim to interact with users and capture their feedback to infer their dynamic preferences, then use the dynamic preferences to make recommendations. Due to its ability to dynamically get the user's feedback, CRS has become an effective solution for capturing dynamic user preferences and solving the explainability problem. CRS can be divided into two main parts of the task: the conversational module for language understanding and generation [14, 19], and the recommendation module for learning strategies for querying and recommendation [16, 18, 20]. This work focuses on the recommendation module.

The Multi-Round Conversational Recommendation (MCR) [5, 16, 18, 32] task is the most realistic assumption in the currently proposed problem scenarios for CRS. In MCR task, the agent continues to interact with the user by asking questions or making recommendations until it successfully meets the user's needs or reaches a maximum number of turns. The main challenge for MCR is how to dynamically learn user preferences and select the right action accordingly, i.e., what attributes to ask for (query action) or what items to recommend (recommend action). CRM [25] and EAR [16] learn user preferences with a factorization-based method under the pairwise Bayesian Personalized Ranking (BPR) framework [23]. SCPR [18] models conversational recommendation as an interactive path reasoning problem on a graph, which is able to prune off many irrelevant candidate attributes. Unicorn [5] builds a weighted graph to model the dynamic preference of users and choose actions from the candidate action space. Despite effectiveness, previous works typically addressed the query action and the recommend action in isolation, without considering their inherent connection. It is crucial to address them jointly for a comprehensive solution.

## 3 Problem Definition

In this section, we formulate the problem of multi-turn conversational recommendation (MCR), which aims to recommend the target item to users by asking for attributes and recommending items in the limited turns of the conversation.

Specifically, we define the user set as $\mathcal{U}$, the item set as $\mathcal{V}$, the attribute set as $\mathcal{P}$, and the set of timestamps as $\mathcal{T} = \{\mathcal{T}_{year}, \mathcal{T}_{month}, \mathcal{T}_{day}, \mathcal{T}_{hour}, \mathcal{T}_{week}\}$. Besides, each item $v \in \mathcal{V}$ is associated with a set of attributes $\mathcal{P}_v \subseteq \mathcal{P}$. And each user $u \in \mathcal{U}$ has a list of items that interacted at different timestamps, denoted by $I_u = \{(v_1, t_1), (v_2, t_2), ..., (v_n, t_n)\}$, where $t_i \in \mathcal{T}$ is a timestamp which can be composed of a list containing different time dimensions, i.e. $t_i = [t_i^y, t_i^m, t_i^d, t_i^h, t_i^{week}]$.

At the beginning of each conversation, the user is initialized with a target item $v_{target}$, an attribute belonging to the target item $p_0 \in \mathcal{P}_{v_{target}}$ and the current timestamp $t_{cur}$. Then, the CRS can ask the user preference on an attribute selected from the candidate attribute set $\mathcal{P}_{cand}$ or recommend a certain number of items from

the candidate item set $\mathcal{V}_{cand}$. Based on the user's feedback, MCR will update the candidate attribute set $\mathcal{P}_{cand}$ and the candidate item set $\mathcal{V}_{cand}$. The conversation will continue until the CRS hits the target item $v_{target}$ or reaches the maximum number of turns $Q$.

## 4 Framework

Each conversational turn of TUT4CRS can be divided into four steps: state modeling, action, transition, and reward.

### 4.1 State Modeling

At first, we define the state $s_q$, which contains all the conversational information of the prior $q$-1 turns: $s_q = [I_u^q, T_{cur}]$, where $I_u^q = [u, \mathcal{P}_{acc}^q, \mathcal{P}_{rej}^q, \mathcal{V}_{rej}^q, \mathcal{P}_{cand}^q, \mathcal{V}_{cand}^q]$ records the user $u$'s dialogue, which includes the accepted attribute set $\mathcal{P}_{acc}^q$, rejected attribute set $\mathcal{P}_{rej}^q$, candidate attribute set $\mathcal{P}_{cand}^q$ and candidate item set $\mathcal{V}_{cand}^q$. $T_{cur} = [t_{cur}^y, t_{cur}^m, t_{cur}^d, t_{cur}^h, t_{cur}^{week}]$ shows the different time dimensions (i.e., year, month, day, hour, week) currently.

### 4.2 Action

According to the current state $s_q$, the agent takes action $a_q \in \mathcal{A}_q$, where $a_q$ can be selected to ask an attribute $p_{ask}^q \in \mathcal{P}_{cand}^q$ or take a recommendation with the items set $\mathcal{V}_{rec}^q \subseteq \mathcal{V}_{cand}^q$. Inspired by [5],

$$\mathcal{V}_{cand}^q = \frac{\mathcal{V}_{\mathcal{P}_{acc}^q}}{(\mathcal{V}_{rej}^q \cup \mathcal{V}_{\mathcal{P}_{rej}^q})}, \qquad \mathcal{P}_{cand}^q = \frac{\mathcal{P}_{\mathcal{V}_{cand}^q}}{(\mathcal{P}_{acc}^q \cup \mathcal{P}_{rej}^q)}, \quad (1)$$

where $\mathcal{V}_{\mathcal{P}_{acc}^q}$ denotes the items that satisfy all the accepted attributes $\mathcal{P}_{acc}^q$, $\mathcal{V}_{\mathcal{P}_{rej}^q}$ denotes the items that satisfy one of the rejected attributes $\mathcal{P}_{rej}^q$, $\mathcal{P}_{\mathcal{V}_{cand}^q}$ denotes the attributes which belong to the candidate items. The details of how the scores are calculated for sorting will be described in section 5.3.

### 4.3 Transition

After the agent has selected an action, the state is updated to the next state $s_{q+1}$ based on the user's feedback. Specifically, when the agent asks for attribute $p_{ask}^q$ and the user accepts it, the accepted attribute set is updated as $\mathcal{P}_{acc}^{q+1} = \mathcal{P}_{acc}^q \cup p_{ask}^q$. Conversely, if the user rejects the attribute, the rejected attribute set is updated as $\mathcal{P}_{rej}^{q+1} = \mathcal{P}_{rej}^q \cup p_{ask}^q$. When the agent decides to recommend the items $\mathcal{V}_{rec}^q$, if the user rejects all the items, the state is updated as $\mathcal{V}_{rej}^{q+1} = \mathcal{V}_{rej}^q \cup \mathcal{V}_{rec}^q$. If the user accepts any of the recommended items, the conversation is successful.

### 4.4 Reward

Following previous CRS study [5, 18], have devised five types of rewards: (1) $r_{rec\_suc}$: Give strongly positive rewards when the user accept recommended items; (2) $r_{rec\_fail}$: Give negative rewards when the user rejects the recommended items; (3) $r_{ask\_suc}$: Give slightly positive rewards when the user accepts the asked attribute; (4) $r_{ask\_fail}$: Give negative rewards when the user rejects the asked attribute; (5) $r_{quit}$: Give strongly negative rewards when the maximum number of turns is reached.

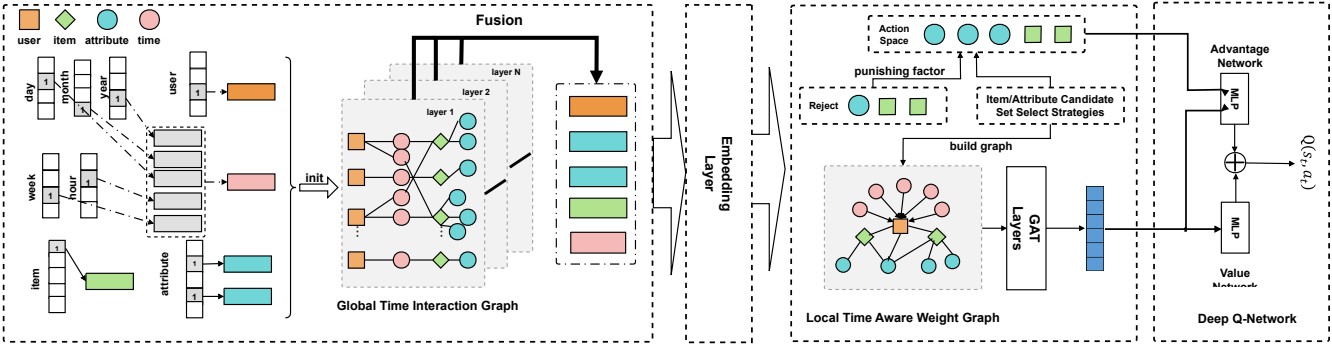

**Figure 2: The overview of TUT4CRS. The left part retains the occasional behaviors in user historical interactions by building a global time interaction graph to learn the rich external information. The middle part is the conversational recommendation module, which leverages the learned external information to construct a local time-aware weight graph, dynamically learns the user's perception of time, and helps make query selections. The right part uses reinforcement learning to decide the next action based on the learned dynamic user preferences.**

## 5 TUT4CRS Policy Learning

The overview of our framework is shown in Figure 2, and we describe this framework in detail in this section.

### 5.1 External Preference Learning

Existing methods [9, 33] model user static preferences from user order history and use this as external information, with unintentional filtering of occasional behavior. However, there is also information implicit in these occasional behaviors, i.e., users' consumption habits at different times, which plays a great role in our subsequent query selection and modeling of user dynamic preference.

However, a small amount of occasional behaviors would be filtered out as noises. To solve this problem and obtain information from occasional behaviors, we use the timestamp as a bridge to construct a triplet relationship $(u, t, v)$. Unlike the binary relationship $(u, v)$, the more complex triple relationship can improve robustness and make each interaction unique, allowing for equal learning of information from them. Based on the triple relationship, we construct a global time interaction graph. With the use of time information as a bridge between the user and the item, we can not only model the user static preferences, but also learn the user's consumption habits at different times, and the item can also gather information about its implied temporal features.

We define the global graph structure as $G_{global} = \{N_{global}, \mathcal{E}_{global}\}$, where $N_{global} = \mathcal{U} \cup \mathcal{V} \cup \mathcal{P} \cup \mathcal{T}$ and $\mathcal{E} = \mathcal{E}_{ut} \cup \mathcal{E}_{tv} \cup \mathcal{E}_{vp}$, i.e., there are edges between user nodes and time nodes, time nodes and item nodes, item nodes and attribute nodes. A time node $t_i \in \mathcal{T}$ represents a specific timestamp, and its embedding is obtained by fusing the embeddings of its different timestamp dimensions as:

$$\begin{cases} e_{t_i}^{(0)} = W_0^T e'_{t_i}, \\ e'_{t_i} = \text{Concat}[t_i^y W_y, t_i^m W_m, t_i^d W_d, t_i^h W_h, t_i^{week} W_{week}], \\ t_i = \{t_i^y, t_i^m, t_i^d, t_i^h, t_i^{week}\}, \end{cases} \quad (2)$$

where each $t_i^{dim} \in t_i$ is the one-hot encode embedding, $dim \in \{y, m, d, h, week\}, W_0 \in \mathbb{R}^{(5*d_{time}) \times d_{hid}}, W_y \in \mathbb{R}^{|\mathcal{T}_{year}| \times d_{time}}, W_m \in \mathbb{R}^{|\mathcal{T}_{month}| \times d_{time}}, W_d \in \mathbb{R}^{|\mathcal{T}_{day}| \times d_{time}}, W_h \in \mathbb{R}^{|\mathcal{T}_{hour}| \times d_{time}}$ and

$W_{week} \in \mathbb{R}^{|\mathcal{T}_{week}| \times d_{time}}$ are trainable parameters. Thus, different time nodes $t_i$ can have different representations.

*5.1.1 Message Propagation and Offline Training.* We employ a $L_{global}$-layer GCN [10–12, 15] to extract the external preferences of users. The initial input embedding of the first layer are $e_u^{(0)}, e_v^{(0)}, e_p^{(0)}$, and $e_t^{(0)}$. Let $e_u^{(l)}, e_v^{(l)}, e_p^{(l)}$, and $e_t^{(l)}$ denote the output embedding of nodes after the propagation of $l$-th layer. For each layer, GCN aggregates [13, 28] neighborhood embedding for each node $n_i$ and combines it with the last layer embedding of the node $n_i$:

$$e_i^{(l+1)} = \text{ReLU}((W^{(l)} \frac{1}{|\mathcal{N}_n|} \sum_{j \in \mathcal{N}_n} e_j^{(l)}) + B^{(l)} e_i^{(l)}). \quad (3)$$

The final embedding of node $n_i$ is as follows:

$$e_i = \text{Mean}([e_i^{(0)}, e_i^{(1)}, ..., e_i^{(L_{global})}]). \quad (4)$$

The similarity between the two representations can be determined through the vector dot product, denoted as $f(e_a, e_b) = e_a^T e_b$. Additionally, the similarity score between an instance $i$ and a set of instances $S$ can be calculated as $g(i, S) = \tanh(\sum_{j \in S} f(e_i, e_j))$. To get more external information from the global graph $G_{global}$, inspired by [9], we use multi-tasks to train the representation offline and obtain external information from different perspectives.

*Attribute-Item Classification Task.* Through this task, we can strengthen the correlation between items and their attributes. For each attribute $p_i$, we classify the items that contain the attribute $p_i$ belonging to the positive class $v_i$ and the items that do not belong to the negative class $v'_i$. Then we strengthen the relation between by using the cross-entropy loss function:

$$\mathcal{L}_{pv} = \sum_{(p_i, v_i, v'_i) \in \mathcal{D}_1} -ln^{f(e_{p_i}, e_{v_i})} - ln^{(1-f(e_{p_i}, e_{v'_i}))}, \quad (5)$$

where the training data $\mathcal{D}_1 = \{(p_i, v_i, v'_i) | v_i \in \mathcal{V}_{p_i}, v'_i \in \mathcal{V}/\mathcal{V}_{p_i}\}$. The attribute $p_i$ is a sample from $\mathcal{P}$, $V_{p_i}$ denotes the positive item set contains the attribute $p_i$ and $\mathcal{V}/V_{p_i}$ denotes the negative item set doesn't contains the attribute.

*User-Time-Item Prediction Task.* For users' consumption habits at different times, we obtain coarse-grained preference information by predicting users' target items under different times, while the items learn their hidden temporal features. Specifically, for each user, his interaction occurs at time $t = \{t^y, t^m, t^d, t^h, t^{week}\}$, and we incorporate the time information of the user:

$$e_u^{time} = W_{fusion}^T \text{Concat}[e_u, e_{time}], \qquad (6)$$

where $W_{fusion} \in \mathbb{R}^{2d_{hid} \times d_{hid}}$ is trainable parameter, and $e_{time}$ is computed by Equation (2). After that, we predict how likely user $u$ will like item $v$ at time $t$ in the conversation state $S^q = \{u, t, \mathcal{P}_{acc}^q, \mathcal{P}_{rej}^q\}$ by:

$$h(v|S^q) = f(e_u^{time}, e_v) + g(v, \mathcal{P}_{acc}^q) - g(v, \mathcal{P}_{rej}^q). \qquad (7)$$

Inspired by [9, 16], we adopt two types of BPR [23] loss with two types of negative examples. The loss function is defined as:

$$\mathcal{L}_{uv} = \sum_{(u,t,v,v^-) \in \mathcal{D}_2} -ln\sigma(h(v|S^q) - h(v^-|S^q))$$
$$+ \sum_{(u,t,v,v^-) \in \mathcal{D}_3} -ln\sigma(h(v|S^q) - h(v^-|S^q)), \qquad (8)$$

where $D_2 = \{(u, t, v, v^-)|v' \in \mathcal{V}/\mathcal{V}_u\}$, the item $v$ is the target item and the $v^-$ is sampled from the set of non-interacted items of user $u$, and $\mathcal{V}_u$ is the set of items historically interacted by user. Also $D_3 = \{(u, t, v, v^-)|v' \in \mathcal{V}_{cand}^q/\mathcal{V}_u\}$, where $\mathcal{V}_{cand}^q$ is the candidate item set that satisfy $S^q$.

*User-Time-Attribute Prediction Task.* In order to obtain users' fine-grained consumption habits at different times, we model by predicting users' attribute preferences at different times. Similar to the previous section, in the conversation state $S^q = \{u, t, \mathcal{P}_{acc}^q, \mathcal{P}_{rej}^q\}$, we predict how likely user $u$ will like attribute $p$ following:

$$\widetilde{h}(p|S^q) = f(e_u^{time}, e_p) + g(p, \mathcal{P}_{acc}^q) - g(p, \mathcal{P}_{rej}^q), \qquad (9)$$

we also employ BPR loss as:

$$\mathcal{L}_{up} = \sum_{(u,t,p,p^-) \in \mathcal{D}_4} -ln\sigma(\widetilde{h}(p|S^q) - \widetilde{h}(p^-|S^q)), \qquad (10)$$

where $\mathcal{D}_4 = \{(u, t, p, p^-)|p \in \mathcal{P}_v, p^- \in \mathcal{P}/\mathcal{P}_v\}$ and $\mathcal{P}_v$ denotes the attribute set belong to the target item $v$.

Finally, the multi-task training objective is:

$$\mathcal{L}_{global} = \mathcal{L}_{pv} + \mathcal{L}_{uv} + \mathcal{L}_{up} + \lambda||\Theta||^2, \qquad (11)$$

where $||\Theta||^2$ is the regularizer term to avoid overfitting, and $\lambda$ is the regularization parameter.

Through the offline training, we can obtain rich external preferences that include user static preferences, user consumption habits, and hidden temporal features of items.

## 5.2 Dynamic Preference Learning

With the approach in the previous section, we captured the rich external preference of the user, which can be of great help to us in the selection of queries and the modeling the user dynamic preferences. To select useful information for the current state from the user external preferences to help us with query selection and user dynamic preference modeling, we construct a local time-aware weight graph $G_{local} = \{N_{local}, \mathcal{E}_{local}\}$ at each turn to analyze what

the user is currently most interested in, where $N_{local} = u \cup t_{cur} \cup V_{cand}^q \cup P_{acc}^q \cup P_{cand}^q$ and $t_{cur} = \{t_{cur}^y, t_{cur}^m, t_{cur}^d, t_{cur}^h, t_{cur}^{week}\}$. Besides, the edge set is following:

$$\mathcal{E}_{local} = \mathcal{E}_{ut} \cup \mathcal{E}_{up_{acc}} \cup \mathcal{E}_{uv_{cand}} \cup \mathcal{E}_{v_{cand}p_{cand}}. \qquad (12)$$

The edge $(u, t) \in \mathcal{E}_{ut}$ denotes the conversation happening in $t \in t_{cur}$; $(u, p) \in \mathcal{E}_{up_{acc}}$ denotes the user likes attribute $p$; $(u, v) \in \mathcal{E}_{uv_{cand}}$ denotes the user may like the item, and $(v, p) \in \mathcal{E}_{v_{cand}p_{cand}}$ denotes the item contains attribute $p$.

To dynamically sense the user's preference, i.e., what exactly the user is interested in, we use GAT [27] to dynamically compute the user's attention during message propagation by:

$$z_i^{(l+1)} = \sigma(\sum_{j \in \mathcal{N}_i} \alpha_{ij} W^{(l)} z_j^{(l)}), \alpha_{ij} = \frac{exp(f(z_i^{(l)}, z_j^{(l)}))}{\sum_{k \in \mathcal{N}_i} exp(f(z_i^{(l)}, z_k^{(l)}))}. \qquad (13)$$

The input representations of user nodes, item nodes, and attribute nodes are obtained through Equation (4), and the representations of time nodes are calculated by the following:

$$z_y^{(0)} = W_{lt}^T(t_{cur}^y W_y), \qquad z_m^{(0)} = W_{lt}^T(t_{cur}^m W_m), \qquad (14)$$
$$z_d^{(0)} = W_{lt}^T(t_{cur}^d W_d), \qquad z_h^{(0)} = W_{lt}^T(t_{cur}^h W_h),$$
$$z_{week}^{(0)} = W_{lt}^T(t_{cur}^{week} W_{week}),$$

where $W_y, W_m, W_d, W_h$, and $W_{week}$ are pretrained by Equation (2) and $W_{lt} \in \mathbb{R}^{d_{time} \times d_{hid}}$ are trainable parameters.

After applying the $L_{local}$-layer GAT, we obtain the final representation $z_i$ of a node $n_i \in \mathcal{N}_{local}$ by using the output $z_i^{(L_{local})}$ of the last layer. The user $u$ utilizes personalized learning to incorporate the current conversation and time information. This enables the user to gather the most relevant and useful information for their current status from external sources. Following [5], we employ a Transformer encoder [26] to capture the sequential information of the conversation history: $\mathbf{X}^* = \text{TransformerEncoder}(\mathbf{X})$, where $\mathbf{X} \in \mathbb{R}^{l \times d_{hid}}$ denotes the embedding of input, and $l$ means the sequence length. The input sequence $\mathbf{X}$ includes the user $u$ and the accepted attributes $P_{acc}^q$ at turn $q$ with the embedding $z_i$ learned by GAT layers. Finally, we get the state representation $s_q$ by a mean pooling layer: $s_q = \text{MeanPool}(\mathbf{X}^*)$.

## 5.3 Action Decision Policy Learning

In order to develop a strategy for decision-making, we employ reinforcement learning [29], which allows us to learn optimal actions. Following [5], we define our action space $\mathcal{A}_q$ as the top-$K_v$ items and top-$K_p$ attributes. To determine the order of the action space at turn $q$, we calculate the *sc* score by:

$$\begin{cases} \text{sc}_p = \sigma\left[e_u^T e_p + h(p, \mathcal{P}_{acc}^q) - h(p, \mathcal{P}_{rej}^q) - h(p, \mathcal{V}_{rej}^q)\right], \\ \text{sc}_v = \sigma\left[e_u^T e_v + h(v, \mathcal{P}_{acc}^q) - h(v, \mathcal{V}_{rej}^q)\right], \end{cases} \qquad (15)$$

where we use user-rejected items as the punishing factor to keep the action in the action space as far away as possible from those items with negative user feedback.

After obtaining the state representation $s_q$ and action space $\mathcal{A}_q$, we use Dueling Q-Network [30] to decide the next action,

which contains two deep neural networks learning the value function $f_V(s)$ and the advantage function $f_A(s, a)$ respectively. The Q-function can be calculated by:

$$Q(s_q, a_q) = f_V(s_q) + f_A(s_q, a_q). \tag{16}$$

Under the standard assumptions of Bellman's equation, the discount factor for delayed rewards is taken to be $\gamma$, and the optimisation objective is to find out the action that achieves the maximum reward by learning the strategy $\pi^*$:

$$Q^*(s_q, a_q) = \mathbb{E}_{s_{q+1}}[r_q + \gamma \max_{a_{q+1} \in \mathcal{A}_{q+1}} Q^*(s_{q+1}, a_{q+1}|s_q, a_q)]. \tag{17}$$

We sample mini-batch experiences from the replay buffer $\mathcal{D}$ and define the loss function as follows:

$$\begin{cases} \mathcal{L}^Q = \mathbb{E}_{(s_q, a_q, r_q, s_{q+1}, A_{q+1}) \sim \mathcal{D}} \left( y_q - Q(s_q, a_q) \right)^2, \\ y_q = r_q + \gamma \max_{aq+1 \in A_{q+1}} Q(s_{q+1}, a_{q+1}). \end{cases} \tag{18}$$

## 6 Experiments

### 6.1 Datasets

We utilize two widely used recommendation datasets for the experimental design. The statistics of them are presented in Table 1. The Yelp dataset is sourced from the 2018 iteration of the Yelp Business Recommendation Challenge. [16] established a hierarchical catalog for Yelp attributes, comprising 29 coarse-grained attributes at the first level and 590 attributes at the second level. We specifically utilize the second level of attributes for items in our study. The MovieLens-20M dataset serves as a prevalent benchmark dataset for recommendation systems. We filter the user-item interactions to include only ratings greater than 3 and subsequently employ $K$-means clustering to categorize attribute types.

**Table 1: Statistics of datasets**

| Dataset | YELP | MovieLens |
|---|---|---|
| #Users | 27,675 | 20,892 |
| #Items | 70,311 | 16,602 |
| #Interactions | 1,368,606 | 2,324,136 |
| #Attributes | 590 | 1,122 |
| #Attribute Types | 29 | 24 |
| #Start Year | 2004 | 1996 |
| #End Year | 2018 | 2015 |

### 6.2 Experimental Settings

*6.2.1 User Simulator.* To facilitate the learning process of the MCR, it is essential to engage in user interactions and receive feedback. To achieve this, we have developed a user simulator that is based on [5, 18]. To simulate a conversation, we consider each observed user-times-item interaction triplet $(u, t, v)$. In this context, the item $v$ serves as the ground-truth, and we define its attribute set $P_v$ as the accepted attribute by the user when asked. The simulated conversation begins with the simulated user specifying a certain attribute $p_0$ randomly selected from $P_v$. During the MCR process, when it comes to selecting the recommendation action, we examine

whether the top-$k_v$ items include the target item $v$. If this is the case, we consider the recommendation accepted by the user and deem the conversation successful. However, if the target item is not included, we continue the conversation until either success is achieved or the maximum number of turns is reached. By utilizing this user simulator, we can enhance the MCR's ability to learn and improve its performance.

*6.2.2 Parameter Settings.* We sort the dataset by time and then split the dataset by 7:1.5:1.5 for training, validation, and testing. And we set the size $K_p$ of the attribute list and the size $K_v$ of the recommendation list as 10, the maximum turn $Q$ as 15. We train the global graph described in Section 5 with the training set for 2500 epochs, the learning rate is 1e-4, the embedding size of attributes and items is set to 64, the embedding size for different timestamp dimensions are set to 16 respectively. And the number of GCN layers $L_{global}$ is set to 2.

We utilize the user simulator to interact with the CRS. This allows us to train our framework effectively using data from the validation set. The settings of the rewards are the same as previous works [5, 33]: $r_{rec\_suc} = 1$, $r_{rec\_fail} = -0.1$, $r_{ask\_suc} = 0.01$, $r_{ask\_fail} = -0.1$, $r_{quit} = -0.3$. The numbers of GAT layers $L_{local}$ for YELP and MovieLens dataset are set to be 4 and 1, respectively. During the training procedure of Dueling Q-Network, the size of the experience replay buffer is 50,000, and the size of the mini-batch is 128. The learning rate and the discount factor $\gamma$ are set to be 1e-4 and 0.999.

*6.2.3 Baselines.* To evaluate the performance of the framework TUT4CRS, we choose four categories of compared models, namely: (1) **Rule-based methods**, i.e., Abs Greedy [3] and Max Entropy [6], which only recommends items to user and selects query based on the maximum entropy; (2) **RL-based SCR (Single-turn Conversational Recommendation) method**, i.e., CRM [25]. A reinforcement learning-based method in SCR scenario, recording the user preferences into a belief tracker; (3) **RL-based MCR method**, i.e., EAR [16], SCPR [18], UNICORN [5]. These methods use reinforcement learning to learn the policy in the MCR setting. In addition, EAR proposes a three-stage solution to enhance the interaction between the conversational component and the recommendation component, SCPR models MCR as an interactive path reasoning problem and UNICORN unifies the two decision strategies via graph neural networks. For a fair comparison, we use FM [22] and TransE [1] to generate initial representations, which we name UNICORN-FM and UNICORN-TransE respectively. Besides, we adjust SCPR (referred to as T*-SCPR) by replacing the FM with the representation learned by Global Graph and concatenate the time representation with the user representation; (4) **RL-based MIMCR (Multi-Interest Multi-round Conversational Recommendation) method**, i.e., MCMIPL [33] and HutCRS [21]. MCMIPL proposes the new conversational recommendation scenario MIMCR, where users may have multiple interests. HutCRS portrays the conversation as a hierarchical interest tree consisting of two stages based on MIMCR.

*6.2.4 Evaluation Metrics.* Following [5], we utilize success rate (SR@Q) to measure the cumulative ratio of successful recommendation with the maximum turn $Q$, and average turn (AT) to evaluate the average number of turns. SR@15 can measure the overall performance of our framework, while SR@10 can represent the successful

**Table 2: Main results. Bold represents the best performance, and underline represents the second-best performance. For SR@10, SR@15, and hDCG, higher values indicate better performance, while for AT, lower values are desirable.**

| Models | YELP | | | | MovieLens | | | |
|---|---|---|---|---|---|---|---|---|
| | SR@ 10 | SR@ 15 | AT | hDCG | SR@ 10 | SR@15 | AT | hDCG |
| Abs Greedy [3] | 0.190 | 0.258 | 13.02 | 0.093 | 0.655 | 0.747 | 7.68 | 0.317 |
| Max Entropy [6] | 0.147 | 0.384 | 13.50 | 0.108 | 0.611 | 0.796 | 9.19 | 0.272 |
| CRM [25] | 0.092 | 0.174 | 14.11 | 0.052 | 0.720 | 0.786 | 6.98 | 0.340 |
| EAR [16] | 0.181 | 0.244 | 13.09 | 0.090 | 0.704 | 0.794 | 7.03 | 0.342 |
| SCPR [18] | 0.244 | 0.479 | 12.66 | 0.141 | 0.620 | 0.854 | 8.71 | 0.304 |
| UNICORN-FM | 0.191 | 0.266 | 12.98 | 0.099 | 0.681 | 0.745 | 7.09 | 0.334 |
| UNICORN-TransE [5] | 0.304 | 0.446 | 11.92 | 0.155 | 0.695 | 0.811 | 7.52 | 0.345 |
| MCMIPL [33] | 0.314 | 0.506 | 12.06 | 0.164 | 0.464 | 0.548 | 10.27 | 0.215 |
| HutCRS [21] | 0.334 | 0.476 | **11.52** | 0.162 | 0.644 | 0.728 | 8.98 | 0.262 |
| T*-SCPR | 0.247 | 0.527 | 12.62 | 0.152 | 0.740 | 0.868 | 7.51 | 0.347 |
| TUT4CRS | **0.341** | **0.541** | 11.77 | **0.167** | **0.821** | **0.894** | **6.63** | **0.352** |

performance of our framework in fewer turns. Besides, hDCG@($Q$, $K$) is used to additionally evaluate the ranking performance of recommendations. For SR@$Q$ and hDCG@($Q$, $K$), higher values represent better performance, while the opposite is true for AT, where lower values represent better.

## 6.3 Performance Comparison

Based on the comprehensive performance analysis presented in Table 2, our TUT4CRS framework demonstrates remarkable effectiveness across both datasets, showcasing superior results in terms of four metrics compared to baselines. Particularly noteworthy are the significant enhancements observed in SR@10 and AT metrics. These improvements can be attributed to several factors: (1) The global graph construction method incorporates a diverse array of external information, encompassing user static preferences, temporal consumption patterns, and item-specific temporal features. These factors collectively contribute to informed query selection and precise modeling of user dynamic preferences. (2) Integration of time information into the modeling of user dynamic preferences, coupled with dynamic learning through attention mechanisms, enables the selection of pertinent information based on current context from previously acquired knowledge. This enables efficient query selection and accurate modeling of user preferences.

Moreover, the utilization of external information proves advantageous in modeling user dynamic preferences. T*-SCPR exhibits notable performance improvements over the original method (i.e., SCPR), particularly evident in SR@10 and AT metrics on the MovieLens dataset. Besides, the choice of initial embedding significantly influences user preference modeling. In the case of UNICORN, we explore two distinct embedding initialization approaches: FM and TransE. Our results reveal that UNICORN-TransE surpasses UNICORN-FM across multiple performance metrics. This underscores the pivotal role of initial embedding quality in harnessing external information effectively, suggesting that incorporating a broader array of diverse external information can yield further enhancements in results.

Furthermore, while MCMIPL and HutCRS demonstrate strong performance on the YELP dataset, their performance on the MovieLens dataset is notably poorer. This variance can be ascribed to the MIMCR scenario inherent in these models, where tasks involve classifying attributes into two distinct levels. However, it's worth noting that while YELP dataset benefits from expert categorization, the MovieLens dataset relies on a clustering algorithm, leading to inferior results. This underscores the significance of expert knowledge in attaining superior performance in the MIMCR scenario, which may not always be readily available in practical applications.

**Table 3: Results of the ablation study.**

| Models | YELP | | | MovieLens | | |
|---|---|---|---|---|---|---|
| | SR@10 | SR@15 | AT | SR@10 | SR@15 | AT |
| OURS | **0.341** | **0.541** | **11.77** | **0.821** | **0.894** | **6.63** |
| instead Random | 0.039 | 0.067 | 14.56 | 0.486 | 0.578 | 9.80 |
| instead FM | 0.074 | 0.106 | 14.24 | 0.490 | 0.565 | 9.69 |
| local w/o GAT | 0.262 | 0.374 | 12.54 | 0.725 | 0.831 | 7.54 |
| local w/o Transformer | 0.131 | 0.321 | 13.62 | 0.696 | 0.802 | 7.57 |
| local w/o timestamps | 0.289 | 0.421 | 12.22 | 0.788 | 0.877 | 6.94 |
| w/o timestamps | 0.289 | 0.460 | 12.15 | 0.780 | 0.880 | 6.92 |

## 6.4 Ablation Studies

In order to verify the effectiveness of some key designs, we conduct a series of ablation experiments on the YELP and MovieLens datasets. The results are shown in Table 3.

*6.4.1 External Information.* In the section 5, we describe the process of constructing a global graph to extract external information from user order history. This information includes user static preference, user consumption habits, and item temporal features. To evaluate the effectiveness of our global time interaction graph mining information, we replace it with other methods, i.e., Random and FM [22]. Interestingly, we observe a significant drop in the performance of the model when these alternative methods are used

instead. This finding highlights the importance and effectiveness of our approach in constructing a global graph.

*6.4.2 Local Graph Components.* Here, we remove GAT, transformer encoder, and timestamp from the local graph respectively. We observe that: (1) When replacing the GAT, we notice a decline in the performance of TUT4CRS. We believe that this is due to the significance of higher-order relationships in effectively modeling user preferences for CRS. (2) Similarly, when the transformer encoder is replaced, TUT4CRS experiences a degradation in performance. This can be attributed to the transformer's strong ability to summarize implicit information within a sequence. (3) The impact of removing timestamps is more pronounced in the YELP dataset compared to the MovieLens dataset. This suggests that the external information we have learned is particularly beneficial for smaller datasets. However, for larger datasets, it is crucial to carefully select the external information that is most relevant to our current state.

*6.4.3 The Role of Timestamp.* To verify the validity of time information, we remove the timestamp entirely from our framework. We observe a decrease in performance on both datasets, underscoring the crucial role of integrating time information into our framework.

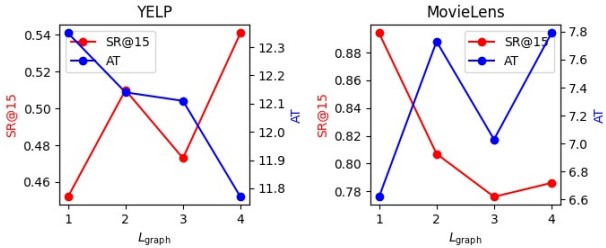

**Figure 3: Impact of the number of GAT Layer($L_{local}$)**

## 6.5 Hyper-Parameter Analysis

By stacking more layers, collaborative information from multihop neighbors would be distilled. We investigate how the number of local time weight graph layer $L_{local}$ influences the performance of TUT4CRS. The range of $L_{local}$ tested here is $\{1, 2, 3, 4\}$, and the results are shown in Figure 3.For YELP dataset, we observe that the performance of the model improves further as the number of layers increases. This would be attributed to the larger size of the local graph, which allows for more effective information to be obtained with deeper layers. For MovieLens dataset, a single layer achieves the best performance, and deeper layers lead to a decrease in performance. It may be because the size of the local graph is small, and deep layers may introduce over-smoothing problems.

## 6.6 Case Study

We conduct a case study on the YELP dataset, focusing on the UNICORN and TUT4CRS models to analyze their performance. Figure 4a shows the detailed MCR process for both models, while Figure 4b displays the attention scores of the user during the first three turns of the TUT4CRS model across different time dimensions.

In this case study, the user initiates a conversation on February 14, 2018, seeking dining options, with the target item being "Choolaah BBQ". In conversation (a), the user's ordering history indicates

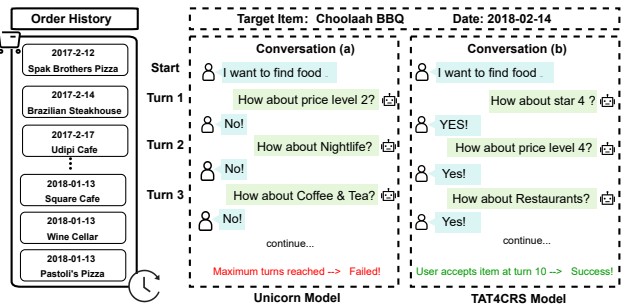

**(a) Detailed MCR process for TUT4CRS and UNICORN. The "price level" refers to the price range of the item, which will be different according to different datasets; "star level" is similar, which refers to the overall rating of the item.**

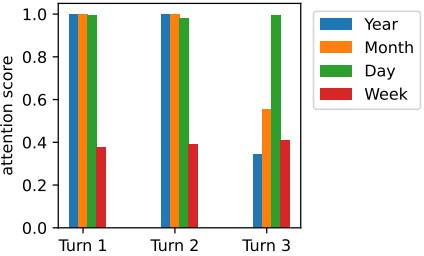

**(b) Users' attention scores for different time dimensions in each turn.**

**Figure 4: The process of TUT4CRS and UNICORN in YELP.**

a preference for pizza places or coffee shops. Consequently, the agent initially queries attributes such as "price level 2", "Nightlife", and "Coffee & Tea". However, these queries are rejected as they are unrelated to the target item. In conversation (b), the user's attention scores for the year, month, and day are higher than the week during turns 1 and 2. Given that the user consumed steak on February 14th of the previous year, the agent infers the user's consumption habits at specific times. Consequently, the agent queries attributes such as "star 4" and "price level 4". In turn 3, the focus shifts to the "day", and the attribute "restaurants" is selected for querying. Ultimately, the recommendation provided by the TUT4CRS model proves successful within ten turns.

## 7 Conclusion

This work presents a novel framework called Time-aware User-preference Tracking for Conversational Recommendation Systems (TUT4CRS). The framework addresses both quantitative and qualitative challenges in CRS by incorporating the time factor as a crucial element. TUT4CRS utilizes a global time interaction graph to capture valuable external information from the user's historical interactions. This information is then used to construct a local time-aware weight graph and personalize the use of external information based on the user current state, which helps the agent to select relevant queries. In this way, the user's dynamic preferences can be learned efficiently and accurately. The effectiveness of TUT4CRS is validated through extensive experiments conducted on two datasets.

## Acknowledgments

This work is supported by National Key Research and Development Program of China (2023YFC3304503), National Natural Science Foundation of China (62276187, 62302333) and Hebei Natural Science Foundation (F2024202047).

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
