# OpenReview forum: "TUT4CRS: Time-aware User-preference Tracking for Conversational Recommendation System"
_acmmm.org/ACMMM/2024/Conference — MM2024 Poster_

### Official Review · Reviewer_Jf9n · 2024-05-24

**Rating:** 6
**Confidence:** 3

**Summary:**

This paper presents a novel framework Time-aware User-preference Tracking for Conversational Recommendation Systems (TUT4CRS). The framework addresses both quantitative and qualitative challenges in CRS by incorporating the time factor. The authors utilize a global time interaction graph to capture external information from the user’s historical interactions, which is used to construct the local time-aware weight graph. This graph then helps to select relevant queries based on the users’ current state. Finally, the effectiveness of TUT4CRS is validated via extensive experiments.

**Strengths:**

This paper makes a good contribution to the CRS which uses the simple yet effective temporal factor to handle the quantity and quality problems in CRS. The proposed framework shows great potential in improving a more precise depiction of user dynamic preferences in a shorter amount of time, and therefore improving the performance of CRS.

**Limitations:**

I do not see the obvious weakness of the paper. I am just curious what is the computation complexity of the current framework (i.e., the training and testing time) and how the proposed method scales with an increasing number of users and interactions.

**Suitability:**

2

---

### Official Review · Reviewer_M3Ln · 2024-05-25

**Rating:** 2
**Confidence:** 3

**Summary:**

The paper presents a novel framework for Conversational Recommendation Systems (CRS) called Time-aware User-preference Tracking for Conversational Recommendation System (TUT4CRS). Traditional CRS face challenges in capturing dynamic user preferences due to limitations in conversation length (quantity) and relevance of queries (quality). TUT4CRS addresses these interconnected issues by leveraging temporal dynamics. It constructs a global time interaction graph to incorporate rich external information and a local time-aware weight graph to dynamically select queries and model user preferences effectively.

**Strengths:**

The paper tackles a compelling and complex problem of accurately modeling dynamic user preferences in conversational recommendation systems.

**Limitations:**

- In problem definition, a set of timestamps is defined as $\mathcal{T} = \mathcal{T}_{𝑦𝑒𝑎𝑟}, \mathcal{T}_{𝑚𝑜𝑛𝑡ℎ},\mathcal{T}_{𝑑𝑎𝑦},\mathcal{T}_{ℎ𝑜𝑢𝑟}, \mathcal{T}_{𝑤𝑒𝑒𝑘}$. Why do we need $\mathcal{T}_{𝑤𝑒𝑒𝑘}}$? Isn’t sufficient to just use  $\mathcal{T} = {\mathcal{T}_{𝑦𝑒𝑎𝑟}, \mathcal{A}_{𝑚𝑜𝑛𝑡ℎ}, \mathcal{A}_{𝑑𝑎𝑦}, \mathcal{T}_{ℎ𝑜𝑢𝑟}$? Also, authors stated that at the beginning of each conversation, the user is initialized with a target item $𝑣_{𝑡𝑎𝑟𝑔𝑒𝑡}$. How is it defined? It’s unclear at this stage.
- The proposed TUT4CRS framework is quite complex, involving multiple components like the global time interaction graph and local time-aware weight graph. This complexity could pose challenges in implementation and require significant computational resources.
- The notation used in the paper is overly complex and extensive, which significantly impacts its readability. The excessive use of symbols, multiple layers of subscripted and superscripted variables, and intricate mathematical expressions make it challenging for readers to follow the core ideas and methodologies. This level of detail, while aiming for precision, can obscure the main contributions and makes comprehension difficult. Simplifying the notation and providing summary tables or diagrams could greatly enhance clarity, making the paper more accessible to a broader audience, including those less familiar with the specific technical nuances.
- The experiments are conducted on only two datasets (YELP and MovieLens). Additional datasets from different domains would help generalize the findings and validate the robustness of the approach.

**Suitability:**

1

---

### Official Review · Reviewer_gzAM · 2024-05-25

**Rating:** 2
**Confidence:** 4

**Summary:**

The authors focus on the challenges in Conversational Recommendation Systems. To better leverage the temporal information, they propose a Time-aware User-preference Tracking method to model the dynamic preference. The framework constructs a global time interaction graph to incorporate rich external information for offline training. During conversational recommendation,
a local time-aware weight graph is constructed to model user dynamic preferences. They conduct experiments on two real-world datasets with addtional analysis.

**Strengths:**

1.The introduction of timestamp information into CRS is a reasonable idea.
2.The paper is well-organized and easy to follow.

**Limitations:**

1.The novelty of the proposed method appears limited. The primary difference from existing methods, such as UNICON, is the incorporation of time embeddings. The other parts such as the graph model in offline training, graph with transformer in CRS and RL are the same. Other components, such as the graph model in offline training, the graph with transformer in CRS, and reinforcement learning , are similar.While this is a useful enhancement, it may not represent a significant departure from existing approaches.
2.It would be beneficial to see how TUT4CRS performs against other baselines that also utilize offline training with timestamp information.
3.There is inconsistency in the model names used throughout the paper, with different names such as TUT4CRS, HUT4CRS, and TAT4CRS

**Suitability:**

1

---

### Official Review · Reviewer_Zbcr · 2024-05-27

**Rating:** 4
**Confidence:** 3

**Summary:**

The article introduces TUT4CRS (Time-aware User-preference Tracking for Conversational Recommendation System), a framework aimed at enhancing Conversational Recommendation Systems (CRS) by leveraging temporal dynamics to model user preferences. The authors propose a method that integrates a global time interaction graph and a local time-aware weight graph to address the limitations of conversational recommendations, specifically the issues of limited conversation turns and low query compliance. Through extensive experiments on two real-world datasets, the framework demonstrates significant improvements in recommendation performance and efficiency.

**Strengths:**

1. Innovative Use of Temporal Dynamics: The incorporation of time as a critical factor in modeling user preferences is a novel approach that bridges the gap between conversation quantity and quality. This adds a valuable dimension to understanding and predicting user behavior.

2. Comprehensive Framework: The dual-layer approach of using both global and local graphs to capture user preferences provides a robust mechanism for handling both long-term and short-term user interests, which enhances the adaptability and accuracy of the recommendations.

3. Detailed Problem Analysis: The paper offers a thorough analysis of the challenges in CRS, particularly the issues arising from limited conversation turns and query compliance. This depth of analysis provides a strong foundation for the proposed solution.

**Limitations:**

1. Figure 3 will appear blurry when enlarged.
2. Are you sure that the two commonly used datasets for recommendation tasks can adequately support conversational recommendation tasks?

**Suitability:**

2

---

### Meta-Review · Area_Chair_LyA7 · 2024-07-02

**Recommendation:** Accept (Poster)
**Confidence:** 4

**Metareview:**

This paper introduces the Time-aware User-preference Tracking for Conversational Recommendation Systems (TUT4CRS) framework, which incorporates time dynamics into CRS to enhance user interaction models. The use of a global time interaction graph and a local time-aware weight graph is commendable and validated through extensive experiments on two datasets. Despite criticisms regarding the limited novelty beyond time embeddings and the complexity of its implementation, the approach marks a step forward in the field. Concerns about consistency in model naming and notation complexity should be addressed to improve clarity.
Anyway, the strengths and potential impact of TUT4CRS justify its acceptance.